# Specific Activation of T Cells by an ACE2-Based CAR-Like Receptor upon Recognition of SARS-CoV-2 Spike Protein

**DOI:** 10.3390/ijms24087641

**Published:** 2023-04-21

**Authors:** Pablo Gonzalez-Garcia, Juan P. Muñoz-Miranda, Ricardo Fernandez-Cisnal, Lucia Olvera, Noelia Moares, Antonio Gabucio, Cecilia Fernandez-Ponce, Francisco Garcia-Cozar

**Affiliations:** 1Institute of Biomedical Research Cadiz (INIBICA), 11009 Cadiz, Spain; pablo.gonzalezgarcia@alum.uca.es (P.G.-G.);; 2Department of Biomedicine, Biotechnology and Public Health, Faculty of Medicine, University of Cadiz, 11002 Cadiz, Spain

**Keywords:** Chimeric Antigen Receptors (CAR), COVID-19, SARS-CoV-2, immunotherapy, CAR-like receptor

## Abstract

Severe Acute Respiratory Syndrome Coronavirus 2 (SARS-CoV-2) is the causative agent of the Coronavirus Disease 2019 (COVID-19) pandemic, which is still a health issue worldwide mostly due to a high rate of contagiousness conferred by the high-affinity binding between cell viral receptors, Angiotensin-Converting Enzyme 2 (ACE2) and SARS-CoV-2 Spike protein. Therapies have been developed that rely on the use of antibodies or the induction of their production (vaccination), but despite vaccination being still largely protective, the efficacy of antibody-based therapies wanes with the advent of new viral variants. Chimeric Antigen Receptor (CAR) therapy has shown promise for tumors and has also been proposed for COVID-19 treatment, but as recognition of CARs still relies on antibody-derived sequences, they will still be hampered by the high evasion capacity of the virus. In this manuscript, we show the results from CAR-like constructs with a recognition domain based on the ACE2 viral receptor, whose ability to bind the virus will not wane, as Spike/ACE2 interaction is pivotal for viral entry. Moreover, we have developed a CAR construct based on an affinity-optimized ACE2 and showed that both wild-type and affinity-optimized ACE2 CARs drive activation of a T cell line in response to SARS-CoV-2 Spike protein expressed on a pulmonary cell line. Our work sets the stage for the development of CAR-like constructs against infectious agents that would not be affected by viral escape mutations and could be developed as soon as the receptor is identified.

## 1. Introduction

Severe Acute Respiratory Syndrome Coronavirus 2 (SARS-CoV-2), an enveloped virus of the Coronaviridae family, emerged in December 2020 and was rapidly propagated all over the world, causing the Coronavirus Disease 2019 (COVID-19) pandemic, which resulted in more than 640,000,000 confirmed cases and 6,600,000 confirmed deaths as of December 2022 (World Health Organization (WHO)). Even though vaccination has greatly tackled the pandemic, recent outbursts in China, where millions are expected to be infected, highlight the need for new tools to face this and other pandemics that may arise in the future.

SARS-CoV-2 harbors a single-strand, positive-sense RNA genome consisting of nearly 30 kpb, which codes for structural proteins (Spike, Envelope, Nucleocapsid and Membrane), eight accessory proteins (ORF14, 3a, 3b, p6, 7a, 7b, 8b and 9b) and other genes such as ORF1a and ORF1b, which, in turn, code for proteins with several roles in viral replication [1,2,3]. Upon infection (mainly through the respiratory tract), SARS-CoV-2 binds the Angiotensin-Converting Enzyme 2 (ACE2), its principal transmembrane receptor, through the Receptor-Binding Domain (RBD) of the Spike protein [4], owing its great contagiousness to the high affinity of such binding [5]. In order to achieve membrane fusion, Spike proteins are cleaved by the Transmembrane Serine Protease 2 (TMPRRS2), also expressed in the host cell surface [3]. Spike proteins have also been shown to be expressed on the surface of infected cells, where TMPRRS2 has been shown to promote syncytia formation among ACE2-positive and Spike-expressing cells, a process that has also been proposed to take place in vivo [6].

Therefore, the SARS-CoV-2 tropism lies mainly in ACE2+ TMPRSS2+ cells, which is an abundant phenotype among epithelial cells of the respiratory tract, such as multiciliate cells of the nasal respiratory epithelium and type II alveolar pneumocytes [7,8]. In addition, Neuropilin-1, a highly expressed protein in both olfactory and respiratory epitheliums, has also been reported as an alternative receptor for SARS-CoV-2 [9]. The virus can replicate at both upper and lower respiratory tracts, promoting an inflammatory state, that in the context of a normal immune response, eventually leads to viral clearance. However, due to genetic and other host factors, some patients suffer from an impaired immune response than includes altered lymphocyte cytolytic activity, inflammatory infiltrates in the lung and an exacerbated release of cytokines, which may lead to multiorgan failure, acute respiratory distress syndrome and macrophage activation syndrome [10,11], thus worsening the disease prognosis [12,13].

It is also important to emphasize that the virus often undergoes mutations that so far have caused higher infectivity. Recently, the genome-wide mutational profile of SARS-CoV-2 was analyzed, revealing that among all its genes, Spike, Nucleocapsid, ORF3a, ORF8, ORF1a and ORF1b are the most susceptible to mutations [14]. However, mutations in the Spike protein are not only responsible for viral stability and transmissibility, as it has been widely reported that emerging Spike variants are associated with both increased infectivity of the virus (higher affinity to ACE2) and improved immune scape, but they also pose a threat to vaccine efficacy [14,15,16]. This is the main reason why variants of concern, such as Delta (B.1.617.2), Omicron (B.1.1.529) and XBB.1 strains, are more likely to elude antibody binding compared to the original strain. Nonetheless, to date, currently available vaccines still provide protection against all variants [17,18,19,20].

Vaccination has been the most efficient means to fight the pandemic. Several vaccines have been approved and commercialized to date, and despite 5,000,000,000 individuals being fully vaccinated three years after the outbreak of the disease (WHO), there are still cases of severe disease due to SARS-CoV-2 infection. Limited vaccination combined with the emergence of new strains due to frequent viral mutations further decreases vaccine efficacy. Furthermore, vaccination may not be an effective option for immunocompromised or aged individuals [21,22].

Other approaches have been assayed, including adoptive cell transfer of immune and stem cells [23], as well as serum from recovered patients or neutralizing monoclonal antibodies (mAbs) [24]. However, antibody-based therapies, while initially effective, can lose effectiveness against new SARS-CoV-2 variants [24,25].

Recently, CAR-T cell therapy, which constituted a breakthrough in immune therapy [26,27,28], has also been used against COVID-19 [29,30,31,32,33]. CARs constitute a tailored molecule that incorporates an MHC-independent recognition domain based on a mAb (a single-chain variable domain (scFv)), a transmission domain that relays activation mimicking the one elicited upon antigen recognition, as well as domains that elicit signals equivalent to those that occur upon activation in an inflammatory scenario, consequently resulting in antigen destruction [34].

In addition to its success in the treatment of hematological tumors, CAR-T cell therapy has also shown promise in non-oncological applications [35], although, as with any mAb-based therapy, its use in acute infections such as COVID-19 would not give a fast enough clinical response, due to the long manufacturing time required and the fact that it would still be subjected to a mutation-derived loss of effectivity [36].

An alternative approach has been explored with the use of the receptor for the virus as the recognition/neutralization domain: hexapeptides corresponding to the ACE2-interacting domain of SARS-CoV-2 have been used to inhibit S1 subunit binding to the ACE2 receptor [37]; ACE2 has also been expressed on the surface of mesenchymal cells [36] and a clinical trial is underway using NKG2D-ACE2 CAR-expressing NK cells [38], although very limited information is available on such a trial.

Here, we propose to use ACE2 as the recognition domain for Spike proteins expressed on infected cells, as it has been shown that ACE2-Fc fusions are more efficient than antibodies at neutralizing different SARS-CoV-2 variants [39]. Moreover, we have used, in addition to the human ACE2 (hACE2) wild-type sequence, a mutant that has been computationally engineered to exhibit increased affinity for the Spike protein [40]. In both ACE2-CAR molecules we have maintained the collectrin-like dimerization domain that has been implicated in facilitating Spike binding [41] and we have introduced an additional mutation that eliminates ACE2 peptidase activity [40]. Using the cellular receptor for the virus as a CAR recognition domain, we establish a proof of concept that would allow for a shorter manufacturing time when new pandemics arise, as it could be designed as soon as the viral entry receptor is identified, without the need for neutralizing antibodies to be isolated and validated from convalescent patients and/or mAb to be developed. Receptor-based recognition domains also eliminate problems derived from mutations, as the virus will still have to recognize its own receptor for cell entry.

## 2. Results

### 2.1. ACE2-CAR-Like Design and Expression

To generate a SARS-CoV-2 Spike recognizing CAR-like synthetic receptor, we fused in-frame hACE2 1–740 residues (containing the 1–17 leader sequence and extracellular domain) to the human IgG_1_ heavy chain (corresponding to the Fc-hinge, CH_2_ and CH_3_ domains—residues 216–446 using Kabat numbering system [42]), followed by the CD28 transmembrane (TM) and intracellular domain and the CD3ζ intracellular domain, yielding the WT-ACE2-CAR construct (Figure 1A,B). From the different possible hinge regions used in CAR constructs (including CD28 (residues 114–152), CD8α (residues 136–182), IgG_4_ (residues 99–327) or just the CH_3_ (residues 221–327)) of IgG_4_, we favored the IgG_1_ hinge region, as ACE2 has already been successfully expressed in this context on mesenchymal cells [36]. The Fc hinge domain was mutated to prevent activation by Fc receptor-expressing cells, thereby minimizing the risk of off-target activation [43].

Alternatively, we generated an affinity-optimized enzymatically inactive construct (AO-ACE2-CAR) (Figure 1C) incorporating mutations that increased affinity to SARS-CoV-2 Spike protein (B.1.1.7 strain) and an additional H345L mutation that eliminates ACE2 peptidase activity, thus avoiding eventual off-target vasodilation effects due to ACE2 conversion of angiotensin II, without affecting the binding affinity for Spike [40].

Both WT-ACE2-CAR and AO-ACE2-CAR were cloned in an HIV-derived lentiviral vector and expressed in Jurkat cells (Figure 1A–C). To easily evaluate CAR-like activation, we used a Jurkat cell line known as Triple Parameter Reporter (TPR) Jurkat, which expresses eGFP, CFP and mCherry, governed by NFAT, NFκB and AP-1 synthetic promoters, respectively [44], and which has previously been validated to evaluate CAR constructs [45,46]. Thus, we transduced Jurkat-TPR cells with the CAR-like constructs and further selected them by means of a co-expressed Blasticidin-S resistance gene. Expression was evaluated with an anti-human IgG antibody that recognizes the hinge region present in both constructs (Figure 1D,E).

As we had the opportunity to compare the performance of CAR-like receptors with different affinities, we first wanted to have equivalent CAR-like expression for both constructs. Taking advantage of anti-IgG staining that equally detected both WT and AO-CAR-like receptors, regardless of their affinity towards Spike, each ACE2-CAR-TPR cell line was sorted into two subpopulations (WT-CAR^High^, WT-CAR^Low^, AO-CAR^High^ and AO-CAR^Low^) depending on their expression levels, matching high-expressing populations in both constructs (Figure 2).

### 2.2. Kinetic of ACE2-CAR Activation upon Stimulation with Spike Protein-Coated Beads

Once both WT-ACE2-CAR- and AO-ACE2-CAR-transduced TPR Jurkat cells were sorted according to CAR expression, we evaluated CAR-mediated activation measuring CD69 upregulation as well as NFAT and NFκB promoter activity upon stimulation with MACSiBeads™ bound to SARS-CoV-2 Spike protein. Figure 3A shows a schematic illustration of the stimulation, FACS histograms are shown in Figure 3B and the corresponding Mean Fluorescence Intensity (MFI) values are plotted together with those obtained upon anti-CD3+ anti-CD28 stimulation in Figure 4. As shown in Figure 3B, CD69 strongly peaks after 24 h post-activation and starts to decline at 48 h, but is still not back at baseline 72 h post-activation. NFAT and NFκB follow similar patterns although with lower intensities. AO-CAR stimulation in all analyzed parameters is similar but stronger than that of WT-CAR (Figure 3B), and differences in intensity are also apparent when stimulations of TRP-Jurkat cells with different levels of CAR expression are analyzed, with a stronger stimulation for cells expressing higher CAR levels.

Subsequently, we wanted to compare CAR-mediated activation with a classical TcR+CD28 activation. Thus, we used the same MACSiBeads™ scaffold, in this case loaded with anti-human-CD3ε and anti-human-CD28 biotinylated antibodies, and plotted the results together with the response corresponding to Spike activation, whose histograms are shown in Figure 3B.

As shown in Figure 4, CD3+CD28-mediated activation is stronger for CD69 and NFAT, while NFκB upregulation is stronger through CAR activation, but only for AO-CAR^High^. Interestingly, CAR activation for AO-CAR^Low^ is similar to WT-CAR^High^, with WT-CAR^Low^ bringing up the rear. Another interesting observation is the fact that CAR-mediated activation lasts longer than CD3+CD28 activation, as CAR activation is still high at 48 h for CD69 and NFAT activation, while CD3+CD28-mediated activation has already decreased at that time point.

Two-way ANOVA assays were performed to compare the intensity of lymphocyte activation elicited by ACE2-CAR, depending on their affinity and level of expression. When comparing against non-stimulated cells, statistical significance is appreciated in every stimulated cell type and at each time point (*** *p* < 0.001) (Figure 4).

If we compare activation between cells expressing the same CAR construct but with different intensity, statistically significant differences were observed between the two WT-ACE2-CAR sorted populations, with activation being stronger in those cells expressing higher levels (WT-CAR^High^). Differences in CD69 expression or NFκB promoter activity were apparent only at 24 h and 48 h (^###^
*p* < 0.001 at both times) after stimulation, while NFAT promoter activity was significantly higher at 24 h, 48 h and 72 h post-stimulation (^###^
*p* < 0.001 at those times). If we consider those cells in which at least one parameter was upregulated (CD69 expression, NFAT promoter activity or NFκB promoter activity), statistically significant differences were observed between those two sorted populations at 24 h (^###^
*p* < 0.001), 48 h (^###^
*p* < 0.001) and 72 h (^#^
*p* < 0.05) after stimulation with Spike protein (Figure 4D). Interestingly, differences were stronger between the two AO-ACE2-CAR sorted populations, being higher in those expressing higher levels (AO-CAR^High^). Thus, CD69 levels and both NFAT and NFκB promoter activity were significatively higher at all time points (24 h, 48 h and 72 h) post-stimulation (^xxx^
*p* < 0.001 at each time point).

When comparing the activation elicited by CARs with different affinities (WT-ACE2-CAR vs. AO-ACE2-CAR), differences were also observed. The AO-CAR^Low^ effector population was significatively higher than WT-CAR^Low^ effector cells, only 24 h and 48 h after Spike stimulation (^###^
*p* < 0.001 at both times), while these differences were wider between AO-CAR^High^ and WT-CAR^High^ cells, which were noticeable 24 h, 48 h and 72 h post-stimulation (^×××^
*p* < 0.001 at each time). Interestingly, no substantial differences were observed between cells expressing lower levels of AO-ACE2-CAR (AO-CAR^Low^) and those expressing higher levels of WT-ACE2-CAR (WT-CAR^High^), confirming that stimulation strength is a result of both affinity and expression level. Unstimulated CAR-expressing cell were not activated when compared to un-transduced cells, indicating that tonic CAR activation is not relevant in our experimental conditions.

### 2.3. Kinetic of ACE2-CAR Activation upon Stimulation with Spike Protein Expressing A549 Cells

We next wanted to evaluate activation of both AO-CAR^High^ and WT-CAR^High^ using Spike expressed on the surface of lung epithelial cells. Thus, A549 cells were transduced with lentiviral vectors harboring the Spike protein (B.1.1.7 strain) coding sequence (Figure 5A). High levels of Spike protein were detected in A549 cells labeled with anti-SARS-CoV-2 Spike RBD mAb, with 98% of the cells being positive for the transgene (Figure 5B).

Both AO-CAR^High^ and WT-CAR^High^ were co-cultured with Spike-A549 cells or un-transduced WT-A549 cells as a control, at different effector:target ratios (Figure 6A). Jurkat cells were identified by their CD69 expression, as A549 cells were negative, while Jurkat cells had a basal expression even in the absence of stimulation (Figure 6B). As shown in Figure 6C, CD69 and both NFAT and NFκB reporters were upregulated in Jurkat cells that had been co-cultured with Spike-expressing A549 target cells.

Consistent with the results obtained in the bead stimulation assay, CD69 upregulation is statistically significant in both AO-CAR- and WT-CAR-expressing cells when compared to non-stimulated TPR cells (*** *p* < 0.001), while AO-CAR-TPR cells exhibited a higher level of activation than WT-ACE2-CAR-TPR cells in terms of CD69 expression, which is significantly higher in AO-ACE2-CAR cells 24, 48 and 72 h after stimulation (^xxx^
*p* < 0.001). The strength of activation is directly dependent on the effector:target ratio, being higher at the 1:2 ratio than at the 1:1 (^###^
*p* < 0.001) and 10:1 (^+++^
*p* < 0.001) ratios for all ACE2-CAR-expressing cells. Moreover, maximum CD69 expression in AO-ACE2-CAR cells occurs 24 h after stimulation, whereas in WT-ACE2-CAR cells, the rate of upregulation is slow and weak but steady, and it is still not downregulated 72 h after stimulation (Figure 7B).

Regarding promoter activity, similar kinetics were observed between NFAT and NFκB. Significant differences were observed between AO- and WT-ACE2-CAR molecules, being significantly higher for AO-ACE2-CAR-TPR cells (^xxx^
*p* < 0.001 for NFAT; ^xx^
*p* < 0.01 for NFκB). As observed for CD69 upregulation, the effector:target ratio also has a great impact on the strength of promoter activity with the maximum average signal for NFAT reporter at the 1:2 ratio, being significantly higher than the 1:1 (^###^
*p* < 0.001) and 10:1 (^+++^
*p* < 0.001) ratios for both AO- and WT-ACE2-CAR-expressing cells (Figure 7A). On the other hand, while the same behavior is observed for the NFκB reporter in AO-ACE2-CAR-TPR cells (Figure 7C), for WT-ACE2-CAR-expressing cells, statistical difference is only found for the 10:1 ratio (^+++^
*p* < 0.001), while there is no statistically significant difference between 1:2 and 1:1 ratios.

It is also noteworthy that for WT-ACE2-CAR-TPR, NFAT and NFκB promoters have still not reached maximal activation 72 h post-stimulation for any effector:target ratio, while for AO-ACE2-CAR-TPR, only cells cultured at a 10:1 ratio are still increasing 72 h post-stimulation. For 1:2 and 1:1 ratios, AO-ACE2-CAR-TPR cells reached maximal activation 48 h after stimulation and declined by 72 h.

Neither WT- nor AO-ACE2-CAR-expressing cells showed signs of activation when co-cultured with un-transduced A549 cells, just as un-transduced TPR cells were not stimulated with A549-Spike-expressing cells.

## 3. Discussion

We used TPR cells to evaluate the functionality of a CAR-like synthetic receptor that recognizes SARS-CoV-2 Spike protein [44]. TPR cells have been shown to be activated by CARs [45,46,47], although less strongly than with phorbol-12-myristate-13-acetate (PMA) and Ionomycin, the tools originally used to select them [44]. Moreover, TPR cells have already been validated and used to evaluate CAR activity [45,46,47]. We have used NFκB and NFAT and not AP-1 reporters, since AP-1 activation can most efficiently be monitor in T cells by the NFAT reporter, as the human distal ARRE-2 site from the IL-2 promoter used as NFAT reporter in TPR cells is dependent on cooperative binding of NFAT and AP-1 and therefore has no activity upon NFAT activation in the absence of AP-1 [48,49,50].

For the initial evaluation of our CAR-like functionality, we have used bead-immobilized Spike trimers, as Lui et al. showed that the trimeric Spike ectodomain binds tightly to ACE2-Fc dimer but not monomers [51], and additionally, Gavriil et al. showed that CAR functionality is largely dependent on hinge-mediated dimerization [52]. Thus, we have recapitulated that scenario by using a CAR-like construct that harbors an ACE2 extracellular domain, hooked up to a Fc hinge, that will display the appropriate dimeric display. Likewise, the construct used for the expression of Spike onto A549 pulmonary cells also display trimeric Spike proteins that have already been shown to properly bind SARS-CoV-2 [53].

We have shown that two synthetic receptors, both based on hACE2 but with different affinities toward Spike, show similar patterns of lymphocyte activation, although with different intensities. We have also compared CAR-mediated activation with that elicited with antibodies against CD28 and CD3ε, which is widely accepted to mimic TcR stimulation [54]. CD3+CD28 co-stimulation also shows similar activation patterns to those of CAR constructs, although with faster activation and fading. It has been widely demonstrated that CAR molecules exhibit a similar function to that of TcR, and it has been accepted as a molecule that mimics TcR activation signals upon antigen binding. However, differences between their signaling kinetics have also been previously described. In this sense, it is worth mentioning that prior studies have shown that both ERK phosphorylation as well as Ca^2+^ signaling are stronger for TcR stimulation, while weaker but more sustained for CAR [55]. As activated ERK promotes downstream signaling cascades that lead to the expression of transcriptional factors such as AP-1 [56] and calcium signaling leads to the activation of both NFκB and NFAT [57,58], the delayed activation elicited by CAR is in keeping with what has been described for ERK and Ca^2+^ signaling. Delayed activation for CARs could also be explained by the need of those molecules to associate with existing TcRs [59,60]. On the other hand, a longer persistence could be explained by a decreased downregulation of CAR expression as opposed to that of TcR/CD3 [61].

CD69 expression is a widely used marker for T cell activation [62] and it has also been extensively utilized to evaluate CAR-mediated activation [63]. CD69 is a transmembrane homodimeric glycoprotein that is rapidly expressed on the T cell membrane upon TcR stimulation [64]. In our study, we have shown that Jurkat cells exhibit their maximum CD69 expression 24 h after CD3+CD28 co-stimulation, decreasing within the following hours, as previously seen in other studies performed in T cells [65]. In contrast, CAR-mediated CD69 upregulation was lower than that mediated by CD3+CD28 co-stimulation 24 h after SARS-CoV-2 Spike stimulation, but was maintained up to 48 h post-stimulation, with the intensity of its expression and the speed of its fading directly depending on both the level of the CAR being expressed and its affinity towards the antigen.

Furthermore, we have tested two CAR-like molecules with different affinities towards SARS-CoV-2 Spike protein, as well as two different levels of CAR-like expression. Previous studies have demonstrated that optimizing the affinity of a second-generation CAR-like molecule against its antigen improves in vivo functionality of CAR-like T cells [66], in agreement with the fact that higher affinities of conventional scFv-based CAR molecules are correlated with higher functionalities [67]. However, higher affinities have exhibited lower discrimination of antigen expression levels while decreasing intercellular contacts, thus lowering exhaustion and increasing other characteristics of the CAR-T cells, such as proliferation and cytotoxicity [67,68,69]. Interestingly, a recent study demonstrated that, similarly to TcR, low-affinity CAR molecules exhibit reduced trogocytosis upon antigen binding in comparison to high-affinity CARs, thus reducing the risk of fratricide cytotoxicity [70]. On the other hand, the density of CAR molecules has also been crucial in CAR-T functionality. A study of T cell signaling performed in Jurkat-TPR cells expressing a CAR against B cell maturation antigen (BCMA) exhibited high levels of activation upon BCMA stimulation in those cells with higher levels of CAR expression, in comparison with those with lower levels [71], which agrees with our results. Our data suggest that, as expected, a higher affinity of the recognition domain of a CAR-like molecule (in this case, AO-ACE-CAR) is translated into a higher lymphocyte stimulation upon antigen binding, a tendency that also occurs among cells expressing higher levels of the CAR molecule.

Both increased affinity and higher expression converge in a higher number of CAR molecules being crosslinked with their antigen, in turn translating into a higher power of the intracellular lymphocyte activation machinery. However, we hypothesize that increments due to affinity vs. expression level occur by different mechanisms. It is presumable that higher affinities are associated with higher binding rates, therefore increasing CAR crosslinking and consequently augmenting the functionality of CAR-T cells, while higher densities could mean a wider accessibility of the cells to their target antigen, therefore recruiting more CAR molecules to the immune synapse and as a result, increasing intracellular signaling pathways. 

The results of CAR-like-derived lymphocyte activation obtained in this study are consistent with those from conventional CAR-T cells, in which their ability to activate and function properly depends on their level of expression on the T cell and their affinity toward the target antigen [72]. Indeed, it has been proposed that there is a CAR affinity threshold, below which lymphocyte activation may be suboptimal [72], which could explain why activation levels remain low without reaching saturation in stimulated WT-ACE2-CAR-TPR cells. In contrast, higher affinities of conventional CAR molecules were also correlated with higher off-target cytotoxicity, as they can recognize cells expressing low levels of the target antigen that could be present in other tissues [69,70]. In our case, this should not be an issue, as SARS-CoV-2 proteins are not expressed naturally in human cells. However, it is also important to mention that higher densities of expression and/or higher affinities to the antigen could cause an exacerbated activation of the CAR-T cell that could lead to an exhausted phenotype [70,71]. In any case, affinity and expression levels should be optimized in clinical scenarios and may even be different among different patients, stressing the importance for an evaluation of both affinity and expression level when assessing a new CAR construct.

Another problem of conventional CAR technologies is that tonic signaling can occur in some CAR-T cells [73]. Although different possible mechanisms have been described, it is worth mentioning that the scFv of conventional CAR molecules can unfold, interacting with domains on adjacent CARs, thus forming oligomerization that can lead to a certain level of activation [74]. Additionally, a conventional CAR containing an unmodified IgG_1_ hinge and a CH_2_-CH_3_ linker was described to bind FcγRI and FcγRII receptors, thus resulting in off-target activation [75]. In our study, we have not reported significant tonic signaling in any of the ACE2-CAR-expressing cells, as we have employed a CAR-like molecule, thus lacking the scFv domain. On the other hand, as the IgG domain that we employed contains a mutation that has been previously described to avoid its recognition by FcR receptors [43], it is expected that IgG-mediated tonic signaling can be also avoided when tested in vivo.

## 4. Materials and Methods

### 4.1. CAR Constructs

WT-ACE2-CAR was engineered with sequences coding for the following residues: 1–740 ACE2, human IgG_1_ heavy chain corresponding to the Fc-hinge, CH_2_ and CH_3_ domains (residues 216–446 using Kabat numbering system [42]), CD28 transmembrane and intracellular domain, and the CD3ζ intracellular domain flanked by attL Gateway™ sequences. Sequences were codon-optimized and synthesized by GeneArt™ (Thermo Fisher Scientific Inc., Carlsbad, CA, USA) and transposed by a LR Gateway™ reaction into the pHR’SINcPPT CEW lentiviral shuttle vector to which an attR flanked Gateway destination cassette had previously been subcloned. The Fc hinge domain was mutated to prevent activation by Fc receptor-expressing cells [43] and CD28 to improve membrane expression [76].

An alternative ACE2-CAR construct was generated (AO-ACE2-CAR), replacing the 1–740 ACE2 coding sequence in the WT-ACE2-CAR by one coding for a computationally engineered, affinity-optimized ACE2 that incorporates mutations described in Glasgow et al. [40] that increased affinity to Spike and an additional mutation that eliminates ACE2 peptidase activity, rendering the construct enzymatically inactive.

### 4.2. SARS-CoV-2 Spike Construct

B.1.1.7 SARS-CoV-2 strain Spike coding sequence was subcloned in the pHR’SINcPPT CEW lentiviral shuttle vector from vector pHDM (NR-53765) containing the SARS-Related Coronavirus 2, Wuhan-Hu-1 Spike Glycoprotein Gene, D614G Mutant with a 21 C-Terminal deletion that improves membrane expression, obtained through BEI Resources, NIAID, NIH.

### 4.3. Cell Lines and Cultures

Human Embryonic Kidney (HEK Lenti-XTM 293T) (Clontech) as well as Jurkat T JE6.1 (Jurkat) and A549 (American Type Culture Collection, Manassas, VA, USA) cell lines were cultured in standard DMEM media supplemented with 10% (*v*/*v*) FBS (Gibco), 1% (*v*/*v*) NEAA, 1% (*v*/*v*) sodium pyruvate, 2 mM L-glutamine, 10 mM HEPES, 100 μg/mL streptomycin, 50 μM 2-mercaptoethanol and 100 units/mL penicillin–streptomycin (Invitrogen, Carlsbad, CA, USA) at 37 °C and 10% CO_2_. Jurkat-TPR are Jurkat T JE6.1 cells genetically modified and established by Prof. Steinberger’s group to express eGFP, CFP and mCherry proteins governed by NFAT, NFκB and AP-1 promoters, respectively [44].

### 4.4. Lentiviral Production and Cell Transduction

HEK Lenti-XTM 293T were used as packaging cell lines to produce lentiviral supernatant. Cells were co-transfected with the corresponding transfer vector in pHRSincPPT-SEW, together with plasmids pCMVΔR8.91, coding for HIV-1 GAG and POL proteins and pMD2.G for the Vesicular Stomatitis Virus G protein (VSVG). Cells were transfected in OptiMEM™ medium (Thermo Fisher Scientific Inc., Carlsbad, CA, USA) by polyethylenimine (PEI)-mediated transfection [77]. Transfection efficiency was evaluated by FACS. Supernatants were collected at 48 and 72 h, centrifuged to remove cells debris and concentrated using Lenti-X™ Concentrator (Clontech) to obtain a high-titer virus-containing pellet. Pellets were subsequently frozen at −80 °C until use. Viral titers were determined, evaluating their efficiency in transducing Jurkat cells. For transduction, cells were plated in fresh media and the day after, lentiviral pellets were thawed resuspended in media and added to the cells. Expression was evaluated 48 h after transduction by FACS.

### 4.5. Flow Cytometry Analysis of Transduced Cells

Expression of both WT-ACE2-CAR and AO-ACE2-CAR were evaluated by staining with a biotinylated Goat anti-Human IgG (Heavy Chain) mAb (antibodies-online, Aachen, Germany), while expression of Spike protein expressed in the A549 cell line was detected by staining with a monoclonal, biotinylated anti-SARS-CoV-2 Spike RBD antibody (Clone 2165) (Leinco, St. Louis, MO, USA), in both cases followed by DyLight™ 650 Neutravidin™ (Thermo Fisher Scientific Inc., Carlsbad, CA, USA) staining. Analyses were performed in a CytoFLEX S V4-B2-Y4-R3 Flow Cytometer (Beckman Coulter Inc., Brea, CA, USA). CD69 expression was analyzed by APC-conjugated anti-human CD69 mAb (Biolegend, San Diego, CA, USA). When indicated, cells were sorted by means of a MoFlo Astrios™ EQ Cell Sorter (Beckman Coulter Inc., Brea, CA, USA).

### 4.6. Jurkat-TPR Stimulation Assays

For bead-immobilized Spike protein, recombinant biotinylated SARS-CoV-2 Spike protein (Recombinant SARS-CoV-2 Spike-Prot (HEK)-Biotin, Miltenyi Biotec, Gaithersburg, MD) was loaded onto anti-Biotin MACSiBead™ Particles (Miltenyi Biotec). Alternatively, biotinylated anti-CD3 and anti-CD28 antibodies (T Cell Activation/Expansion Kit, human, Miltenyi Biotec) were loaded as a control for T cell activation.

Cells (1 × 10^5^) were plated in 96-well flat-bottomed plates in 25 μL supplemented DMEM (see Section 4.3) and the corresponding coated particles (2 × 10^5^ in 25 μL) were subsequently added to the corresponding wells in a bead-to-cell ratio of 1:2. Cells were incubated at the resulting density of 2 × 10^6^ cells/mL, for 20 min at 37 °C and 10% CO_2_ to enhance bead-to-cell interactions, and then diluted with supplemented DMEM to achieve a final cell density of 0.5 × 10^6^ cells/mL. Plates were then incubated at 37 °C and 10% CO_2_, and the activation was analyzed at 3 different time points: 24 h, 48 h and 72 h after stimulation. For each measurement, CD69 staining as well as the signal given by TPR promoter reporters were analyzed.

For cell-mediated stimulation, Spike-expressing A549 target cells were plated and cultured in 96-well plates at 40,000, 20,000 and 2000 cells/well for 1:2, 1:1 and 10:1 efector:target ratios, respectively, and irradiated at 30 Gy in an X-ray irradiator (Faxitron X-ray LLC 43855F-CP160, Lincolnshire, IL, USA). Effector ACE2-CAR-TPR cells were then added at 20,000 cells/well in a total of 200 µL for all ratios tested and cultured for the indicated time periods at 37 °C in a 10% CO_2_ incubator. Activation was also analyzed 24 h, 48 h and 72 h after activation.

### 4.7. Statistical Assays

Statistical analyses were performed using Prism 5.0 GraphPad (GraphPad, La Jolla, CA, USA). Analysis of differences between two groups was performed using an unpaired two-tailed Student’s *t* test, whereas analysis of differences between more than two groups was performed using a two-way ANOVA, followed by the Bonferroni post hoc test. Values are represented as mean from triplicates ± SD, and significance levels are indicated in the figures (* *p* < 0.05; ** *p* < 0.01; *** *p* < 0.001).

## 5. Conclusions

This paper shows the feasibility of a CAR-like construct based on the natural receptor of a pathogen (in this case, ACE2 for SARS-CoV-2), showing that the receptor promotes activation with different levels of stimulation depending on the affinity towards its target and with negligible levels of tonic signaling. Thus, we have established a proof of concept for the development of CAR-like constructs against an infectious agent that will not require monoclonal antibody-derived sequences, which are slower to develop. Severe infectious patients could be subjected to CAR-like based therapies by obtaining T cells and transducing them with already-prepared CAR-like constructs. Taking advantage of the infrastructure already developed to treat cancer patients with CAR-T cells, and following the same procedures, CAR-like T cells can be infused back into the patient to fight virus-infected cells, thus aiding the cellular immunity in those patients where it is ineffective. Alternatively, off-the-shelf allogenic T cells with silenced TcR and MHC could be used as a general-use medicinal product that will not need to be prepared for each patient. An additional advantage of CARs based on natural receptors, as opposed to antibody-derived sequences, would be their imperviousness to pathogen mutations, as pathogen survival is dependent on keeping their binding ability to their entry receptors.

## Figures and Tables

**Figure 1 ijms-24-07641-f001:**
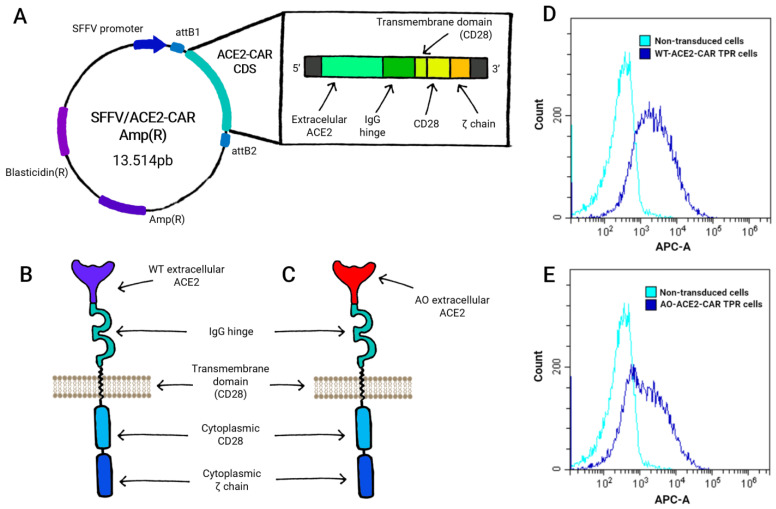
ACE2-CAR design and expression. Generic diagram of the expression vector SFFV/ACE2-CAR Amp(R) (**A**), which was the same for both WT- (**B**) and AO- (**C**) ACE2-CAR molecules, only differing at the ACE2 extracellular domain. Expression of both CAR-like constructs onto Jurkat-TPR cells was detected by staining with anti-IgG, recognizing the hinge region that was unchanged between WT- (**D**) and AO- (**E**) ACE2-CAR.

**Figure 2 ijms-24-07641-f002:**
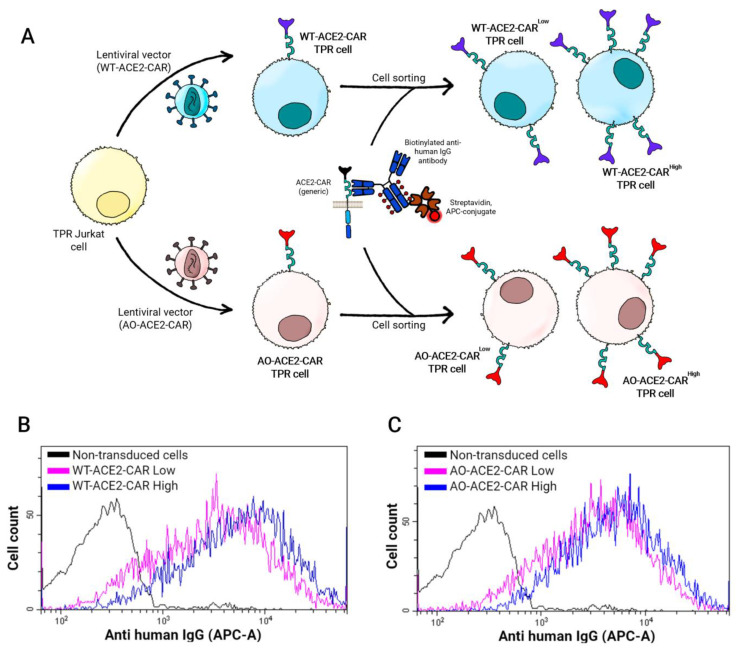
Establishment of ACE2-CAR-expressing Jurkat-TRP cell lines. (**A**) Schematic representation of the sorting procedure: cells were sorted according to anti-IgG staining, recognizing the IgG-derived hinge domain of the ACE2-CAR molecules. The terms “Low” and “High” of the sorted WT (**B**) and AO (**C**) ACE2-CAR-TPR populations refer to lower and higher densities, respectively, of the protein expressed in the cell membrane.

**Figure 3 ijms-24-07641-f003:**
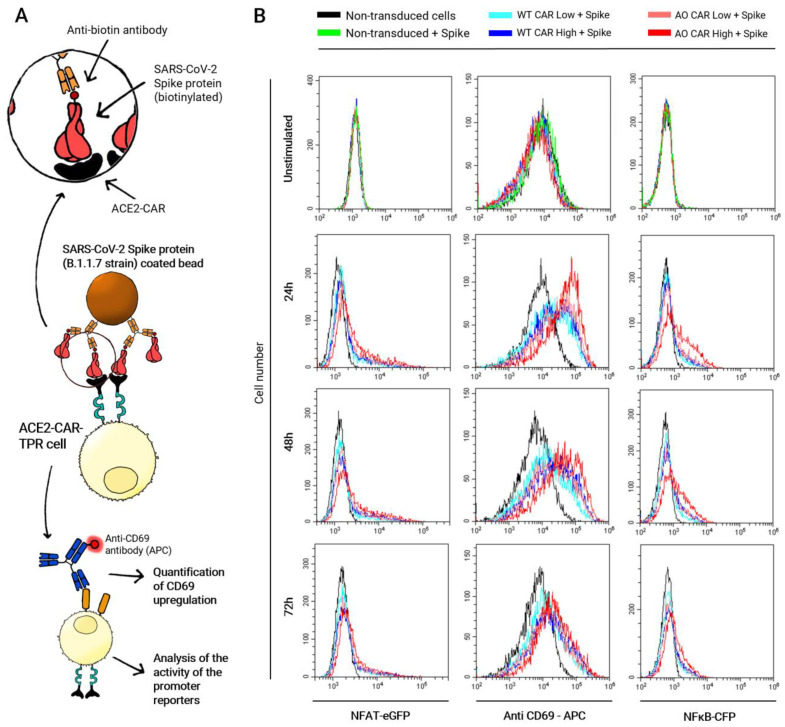
Activation of ACE2-CAR-TPR cells upon stimulation with Spike-coated MACSiBead particles. (**A**) Schematic illustration of the stimulation of ACE2-CAR-TPR cells with Spike and (**B**) Histogram overlays showing eGFP (NFAT), CFP (NFκB) or CD69 upregulation 24, 48 and 72 h after stimulation with Spike trimers. A representative experiment out of three performed in triplicate is shown.

**Figure 4 ijms-24-07641-f004:**
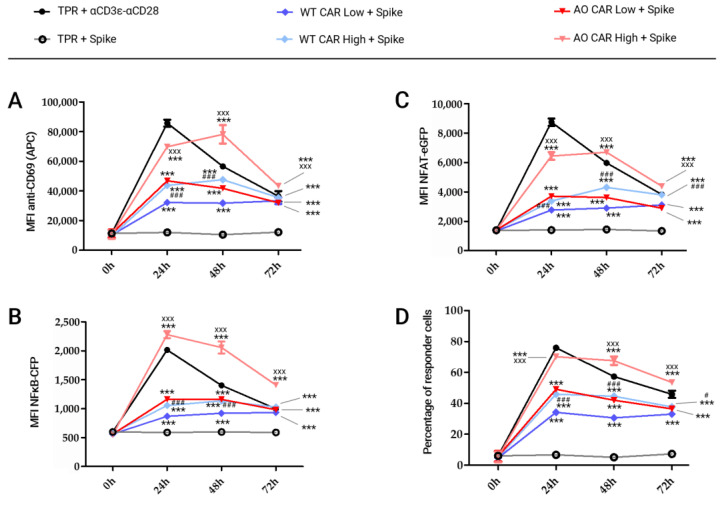
Kinetics of the activation of ACE2-CAR-TPR cells upon stimulation with Spike- or anti-CD3+anti-CD28 MACSiBead-coated particles. MFI values for CD69 upregulation (**A**) as well as NFκB (**B**) or NFAT promoter activation (**C**) upon either Spike trimers or CD3+CD28 stimulation. Percentage of the cell that responded to at least one of the analyzed activation markers: NFAT, NFκB and/or CD69 (**D**). Each symbol represents mean ± SEM of MFI values of three independent experiments performed in triplicate. Statistical analysis is indicated for each condition vs. non-stimulated cells (*** *p* < 0.001), Spike-stimulated WT-CAR^High^ vs. WT-CAR^Low^ (^#^
*p* < 0.05, ^###^
*p* < 0.001) and Spike-stimulated AO-CAR^High^ vs. AO-CAR^Low^ (^xxx^
*p* < 0.001).

**Figure 5 ijms-24-07641-f005:**
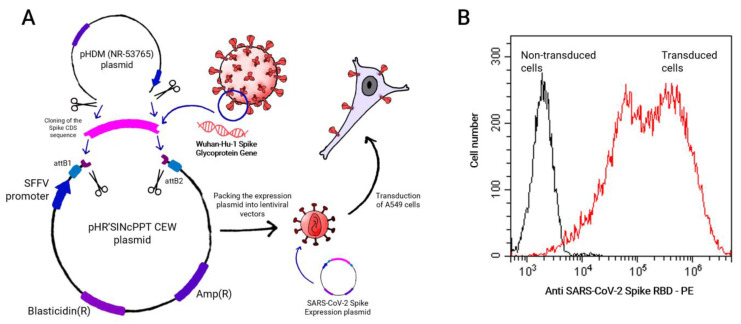
Expression of SARS-CoV-2 spike protein on A549 cells. (**A**) Schematics of SARS-CoV-2 Spike-expressing A549 cells and (**B**) detection, by flow cytometry, of Spike expression of transduced A549 cells.

**Figure 6 ijms-24-07641-f006:**
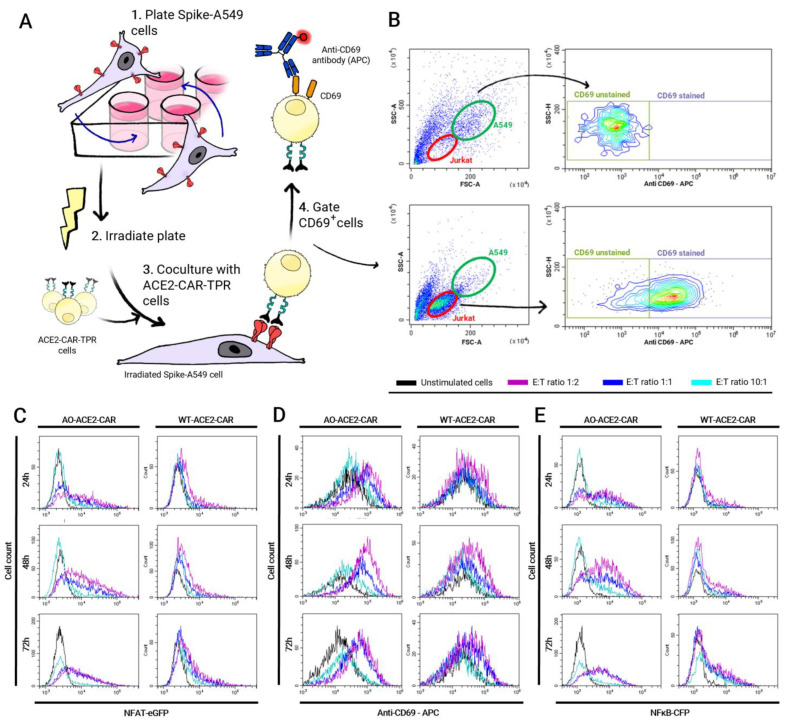
Jurkat-A549 co-culture assay. Schematic representation of the procedure (**A**) and illustration of FACS gating strategy (**B**), which allowed Jurkat (below) to be distinguished from A549 cells (above) based on FSC and SSC as well as on CD69 staining expressed in Jurkat over A549 even in the absence of stimulation. (**C**–**E**) Histogram overlays showing eGFP (NFAT), CFP (NFκB) or CD69 upregulation 24, 48 and 72 h after co-culturing with A549-Spike target cells at 1:1 effector:target ratios, at different time points. A representative experiment out of three performed in triplicate is shown.

**Figure 7 ijms-24-07641-f007:**
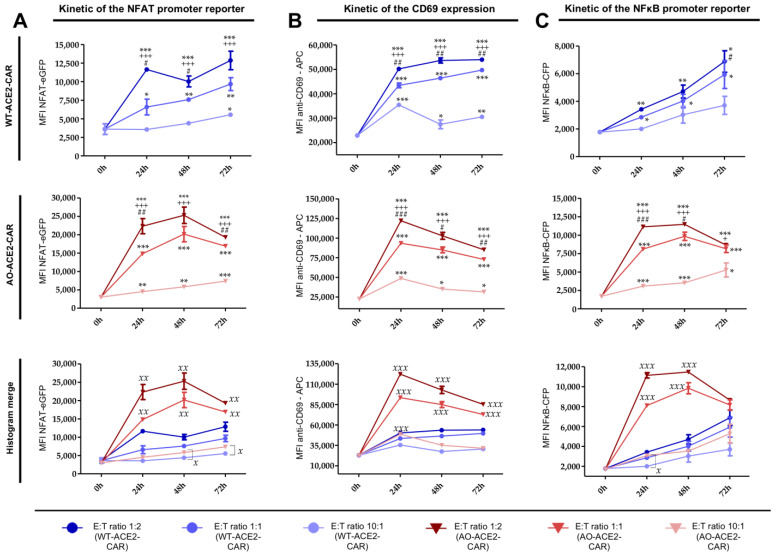
Kinetics of the activation of ACE2-CAR-TPR cells upon culture with Spike-expressing A549 cells. Kinetics of CD69 upregulation (**B**) as well as NFκB (**C**) or NFAT promoter activation (**A**) upon culture with Spike-expressing A549 cells. In the two first rows, MFI values of the corresponding activation markers for WT-CAR-TPR cells (top) and AO-CAR-TPR cells (middle) at different ratios were shown while in the last row, values from the 1:1 effector:target ratios for all cell types are compared. Each symbol represents mean ± SEM of three independent experiments performed in triplicate. Statistical analysis is indicated for each condition vs. non-stimulated cells (* *p* < 0.05, ** *p* < 0.01, *** *p* < 0.001), ratio 2:1 vs. 1:1 for each condition (^#^
*p* < 0.05, ^##^
*p* < 0.01, ^###^
*p* < 0.001); ratio 2:1 vs. 10:1 for each condition (^+^
*p* < 0.05, ^+++^
*p* < 0.001); WT-CAR vs. AO-CAR for each ratio (^x^
*p* < 0.05, ^xx^
*p* < 0.01, ^xxx^
*p* < 0.001).

## Data Availability

Data is contained within the Appendix A.

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
