# Peer review of "Specific Activation of T Cells by an ACE2-Based CAR-Like Receptor upon Recognition of SARS-CoV-2 Spike Protein"

_ijms, 2023, doi:10.3390/ijms24087641_

Round 1

Reviewer 1 Report

The manuscript “Specific Activation of T Cells by an ACE2-based CAR-Like Receptor Upon Recognition of SARS-CoV-2 Spike Protein” is dedicated to developing the chimeric antigen receptor (CAR) therapy to COVID-19. Authors ambitiously work towards possible prospective development of the pandemics by proposing time-effective way to generate Spike-specific antibodies. The research presented in the article opens perspective for a technology that will be of utmost value during the pandemics to come.

However, the authors have missed one important aspect of the SARS-CoV-2 infectivity, namely, the formation of syncytia. The virus entry into ACE2+/TMPRSS2+ cells leads to fusion of cells and formation of polynucleated cells. It has been proposed that such process may take place in vivo (10.15252/embj.2020106267), thus this could be an alternative way of virus resistance to all the prophylactics efforts as the virus becomes unreachable to the antibodies. This notion does not influence the findings of the authors and is merely aimed to bring the whole picture to the introduction.

There couple of general issues. First, that authors describe piece of research but do not provide any evidence of it (Lines 186-189). Either they should note that data are not shown or provide the data as a supplementary material. Second, the authors need to elaborate more on some of the abbreviations. While some of them are wildly known (as CH-domain or NFAT), some are less familiar to the wide scientific community (as MFI, TPR or PMA). I would recommend explaining those latter ones.  

The article is written well, in good English, and presents consistent results. The Introduction is clear and easy to follow. The results are described consistently, totally supporting the conclusions that authors draw. The statistical analysis is performed. 

The minor corrections are as follows:

In Introduction:

Line 109 – “will stablish” probably is typo? Is it “we will establish”?

In Results:

Lines 168, 190, etc. – it seems the authors have missed couple of corrections. I believe IJMS uses “Figure” as it is used throughout the manuscript, but not “figure” as it presents in couple of places.

Lines 142, 159 – I would recommend rephrasing the “respectively to lower and higher densities” to more common “to lower and higher densities, respectively”.

Line 249 – correct to “be distinguished”.

In Discussion:

Line 326 – correct to “Ca2+”.

In Materials and Methods:

Line 412 – “et.al.” in Italic.

Author Response

Dear Sir/madame

Thanks for your thorough review and suggestions. Please find below the answers to each of your comments:

  • However, the authors have missed one important aspect of the SARS-CoV-2 infectivity, namely, the formation of syncytia. (…) This notion does not influence the findings of the authors and is merely aimed to bring the whole picture to the introduction.

We have indeed overlooked this aspect, and we agree that including it gives a broader perspective to the introduction. Thus, we have included a paragraph addressing the issue and the corresponding reference.

“Spike proteins has also been shown to be expressed on the surface of infected cells, where TMPRRS2 has been shown to promote syncytia formation among ACE2 positive and Spike expressing cells, a process that has also been proposed to take place in vivo [6]”

  1. Buchrieser, J.; Dufloo, J.; Hubert, M.; Monel, B.; Planas, D.; Rajah, M. M.; Planchais, C.; Porrot, F.; Guivel‐Benhassine, F.; Werf, S. V. d.; Casartelli, N.; Mouquet, H.; Bruel, T.; Schwartz, O., Syncytia formation by SARS‐CoV‐2‐infected cells.
  • authors describe piece of research but do not provide any evidence of it (Lines 186-189).

We understand that the referee is referring to the following paragraph: “Two-ways ANOVA assays were performed to compare the intensity of lymphocyte activation elicited by ACE2-CAR, depending on their affinity and level of expression. When comparing against non-stimulated cells, statistical significance is appreciated in every stimulated cell type and at each time point (***p<0.001).”

This data refers to Figure 4, and it is actually misplaced, so we have moved the paragraph to the end of the next set of data and refer to Figure 4.

  • the authors need to elaborate more on some of the abbreviations. While some of them are wildly known (as CH-domain or NFAT), some are less familiar to the wide scientific community (as MFI, TPR or PMA).

We have included definition and abbreviations between “()” at their first appearance.

For “TPR”, we had originally addressed it in line 144 together with Jurkat cells “Jurkat cell line known as Triple Parameter Reporter Jurkat (Jurkat-TPR),” but considering we later on for simplicity's sake use TPR abbreviation in the absence of the term Jurkat, we have directly clarified the reference TPR by its own.

  • Line 109 – “will stablish” probably is typo? Is it “we will establish”?

Indeed, it is a typo that actually does not appear in the version that has been sent to us after revision.

 Lines 168, 190, etc. – it seems the authors have missed couple of corrections. I believe IJMS uses “Figure” as it is used throughout the manuscript, but not “figure” as it presents in couple of places.

Indeed, we have identified 7 typos such as the one described, and we have corrected them in the text.

 Lines 142, 159 – I would recommend rephrasing the “respectively to lower and higher densities” to more common “to lower and higher densities, respectively”.

It has been corrected in the text according to your suggestion.

 Line 249 – correct to “be distinguished”.

It has been corrected according to your suggestion.

 Line 326 – correct to “Ca2+”.

It has been corrected according to your suggestion.

  • Line 412 – “et.al.” in Italic.

It has been corrected according to your suggestion.

Reviewer 2 Report

In the presented manuscript, the authors suggested to use CAR technology to develop a drug against SARS-CoV-2. It is proposed to use as the CAR molecule wild-type or affinity-optimized ACE2-Fc fusion construct. Both constructs were expressed in TPR Jurkat cells. The authors showed that the Spike homotrimer immobilized on MACSiBeads™ or expressed in A549 cells causes ACE2-CAR mediated activation of TPR Jurkat cells. The authors believe that their approach will make it possible to create CAR-like constructs against other viruses for which receptors are known.

Comments

To quantify the effect of Spike on ACE2-CAR activity, subpopulations of TPR Jurkat cells with low and high levels of expression of two ACE2-CAR species, wild-type and affinity-optimized WT, were used. What was the amount of ACE2-CAR in a single cell at low and high density?

The article almost does not discuss how ACE2-CAR can be used to treat patients with COVID-19. I would like the authors to discuss this issue in more detail.

Lines 459-461. Particles were added at a density 2x10^5 in 25 ul to 10^5 cells in 25 ul. This seems inconsistent with the statement that “particles were added to the corresponding wells in a bead-to-cell ratio 1:2”

Lines 468-472. In the Methods section, the following numbers of A549 cells are indicated: “40.000, 8.000 and 4.000 cells/well for 1:1, 5:1 and 10:1 efector:target ratios respectively”. It further states that “Responder ACE2-CAR-TPR cells were then added at 40.000 cells/well in a total of 200uL”. This does not match the description of the experiment (lines 259-261), where effector:target ratios are 1:1, 1:2 and 1:10. For a better understanding, it was worth pointing out which cells are designated as effector and which as target.

Author Response

Thanks for your thorough review and suggestions. Please find below the answers to each of your comments:

  • To quantify the effect of Spike on ACE2-CAR activity, subpopulations of TPR Jurkat cells with low and high levels of expression of two ACE2-CAR species, wild-type and affinity-optimized WT, were used. What was the amount of ACE2-CAR in a single cell at low and high density?

The amount of membrane molecules would be useful to compare among different constructs, although this comparison would be complicated anyway due to different signaling domains. Our initial purpose when sorting cell populations expressing different ACE2-CAR density, was primarily to have populations with a similar level of expression (high) so we could compare the effect of ACE2-CAR with different affinities, but then additionally, we had the opportunity to compare different level of expression between cell expressing the same construct. Being the same molecule, we have relied on fold changes just to gauge modulation due to changes in the expression level.

To offer an exact determination of the number of molecules, the most straightforward approach would be to build a standard curve using capture beads with known different Antibody Binding Capacity (ABC) such as the Quantum™ Simply Cellular® (Bangs Laboratories, Inc.) used in [1] or [2]. Mean Fluorescence Intensity (MFI) corresponding to each known ABC value could be obtained, and thus unknown ABC could be resolved from MFI values. Alternatively, a western blot with membrane preparations (or Sulpho-NHS biotinylated cells) could be run in parallel with IgG standard with known quantity (as the hinge region in our ACE2-CAR construct shares epitopes with human IgG where the sequence comes from). As we did not have either Quantum™ Simply Cellular® or human IgG available nor would it be delivered in the 5 days period granted to answer, we built our own standards by incubating VersaComp® Antibody Capture Beads (Beckman Coulter, Inc.) with different concentration of the same antibody used for labeling, under conditions where all antibody molecules would be bound.  We estimate 25000 molecules for CARHigh, while CARLow was in the 11000 range. Those values are a little below those reported by Ho et al. or Rodriguez-Marquez et al. [1, 2] which is in keeping with the fact that they used much stronger promoters such as E1alfa or MNC as opposed to our SFFV.

  • The article almost does not discuss how ACE2-CAR can be used to treat patients with COVID-19. I would like the authors to discuss this issue in more detail.

Indeed, we have not expanded our view on how to treat patients based in our proof of concept, but it would definitely be an interesting addition. We are thus adding a paragraph in the conclusion section:

“Severe infectious patients could be subjected to CAR-like based therapies by obtaining T cells and transducing them with already prepared CAR-like constructs. Taking advantage of the infrastructure already developed to treat cancer patients with CAR-T cells, and following the same procedures, CAR-like-T cells can be infused back to the patient to fight virus infected cells, thus aiding the cellular immunity in those patients where it is ineffective. Alternatively, off the shelf allogenic T cells with silenced TcR and MHC could be used as a general use medicinal product that will not need to be prepared for each patient.”

  • Lines 459-461. Particles were added at a density 2x10^5 in 25 ul to 10^5 cells in 25 ul. This seems inconsistent with the statement that “particles were added to the corresponding wells in a bead-to-cell ratio 1:2”

We mentioned that both cells and particles were added in a 25mL volume just to clarify how the experiment was performed, but the relevant information, was the total number of cells and beads added to each well (1 × 10^5 cells and 2·10^5 particles). The term “density” used in this paragraph is indeed misleading because we are stating total number of beads and cells added to each well, thus we have eliminated the term “density” while keeping it later on where it applies, when we state the density of cells per mL.

The final paragraph would read as follows: “Cells (1 × 10^5) were plated in 96-well flat bottom plates in 25μL supplemented DMEM (see 4.3) and the corresponding coated particles (2·x 10^5 in 25μL) were subsequently added to the corresponding wells in a bead-to-cell ratio 1:2. Cells were incubated at the resulting density of 2 x 10^6 cells/mL, for 20 minutes at 37ºC and 10% CO2 to enhance bead-to-cell interactions, and then diluted with supplemented DMEM to achieve a final cell density of 0.5 x 106 cells/mL.”

 Lines 468-472. In the Methods section, the following numbers of A549 cells are indicated: “40.000, 8.000 and 4.000 cells/well for 1:1, 5:1 and 10:1 efector:target ratios respectively”. It further states that “Responder ACE2-CAR-TPR cells were then added at 40.000 cells/well in a total of 200uL”. This does not match the description of the experiment (lines 259-261), where effector:target ratios are 1:1, 1:2 and 1:10. For a better understanding, it was worth pointing out which cells are designated as effector and which as target.

There is a mistake on our part in the method section. It has been modified and now reads:

“Spike-expressing A549 target cells were plated and cultured o/n in 96well plates at 40000, 20000 and 2000 cells/well for 1:2, 1:1 and 10:1 efector:target ratios respectively and irradiated at 30Gy in a X ray irradiator (Faxitron X-Ray LLC 43855F-CP160, Lincolnshire, IL). Effector ACE2-CAR-TPR cells were then added at 20000 cells/well”

For the description of the experiment, it now reads 1:2, 1:1 and 10:1.

We have also added the term “target” each time we refer to Spike-expressing A549 cells, so now it reads “Spike-expressing A549 target cells” and while previously using the term “responder” or “effector” indistinctly we now use only the term “effector” throughout text and figures.

References

  1. Ho, J.-Y.; Wang, L.; Liu, Y.; Ba, M.; Yang, J.; Zhang, X.; Chen, D.; Lu, P.; Li, J., Promoter usage regulating the surface density of CAR molecules may modulate the kinetics of CAR-T cells in vivo. Mol Ther - Methods Clin Dev 2021, 21, 237-246. 10.1016/j.omtm.2021.03.007 PMID - 33869653
  2. Rodriguez-Marquez, P.; Calleja-Cervantes, M. E.; Serrano, G.; Oliver-Caldes, A.; Palacios-Berraquero, M. L.; Martin-Mallo, A.; Calviño, C.; Español-Rego, M.; Ceballos, C.; Lozano, T.; Martin-Uriz, P. S.; Vilas-Zornoza, A.; Rodriguez-Diaz, S.; Martinez-Turrillas, R.; Jauregui, P.; Alignani, D.; Viguria, M. C.; Redondo, M.; Pascal, M.; Martin-Antonio, B.; Juan, M.; Urbano-Ispizua, A.; Rodriguez-Otero, P.; Alfonso-Pierola, A.; Paiva, B.; Lasarte, J. J.; Inoges, S.; Cerio, A. L.-D. d.; San-Miguel, J.; Larrea, C. F. d.; Hernaez, M.; Rodriguez-Madoz, J. R.; Prosper, F., CAR density influences antitumoral efficacy of BCMA CAR T cells and correlates with clinical outcome. Sci Adv 2022, 8, (39), eabo0514. 10.1126/sciadv.abo0514 PMID - 36179026